# PeerJ

# National estimates of emergency department visits for pediatric severe sepsis in the United States

Sara Singhal[1], Mathias W. Allen[2], John-Ryan McAnnally[3], Kenneth S. Smith[4], John P. Donnelly[5] and Henry E. Wang[6]

[1] Department of Emergency Medicine, University of Kentucky, Lexington, Kentucky, USA
[2] Department of Emergency Medicine, Wake Forest School of Medicine, Winston-Salem, North Carolina, USA
[3] Department of Emergency Medicine, University of Tennessee College of Medicine, Chattanooga, Tennessee, USA
[4] University of Alabama, School of Medicine, USA
[5] Department of Epidemiology, University of Alabama at Birmingham, USA
[6] Department of Emergency Medicine, University of Alabama School of Medicine, USA

Corresponding author
Henry E. Wang, hwang@uabmc.edu

## ABSTRACT

**Objective.** We sought to determine the characteristics of children presenting to United States (US) Emergency Departments (ED) with severe sepsis.

**Study design.** Cross-sectional analysis using data from the National Hospital Ambulatory Medical Care Survey (NHAMCS). Using triage vital signs and ED diagnoses (defined by the International Classification of Diseases, Ninth Revision codes), we identified children <18 years old presenting with both infection (triage fever or ICD-9 infection) and organ dysfunction (triage hypotension or ICD-9 organ dysfunction).

**Results.** Of 28.2 million pediatric patients presenting to US EDs each year, severe sepsis was present in 95,055 (0.34%; 95% CI: 0.29–0.39%). Fever and respiratory infection were the most common indicators of an infection. Hypotension and respiratory failure were the most common indicators of organ dysfunction. Most severe sepsis occurred in children ages 31 days–1 year old (32.1%). Most visits for pediatric severe sepsis occurred during winter months (37.4%), and only 11.1% of patients arrived at the ED by ambulance. Over half of severe sepsis cases were self-pay or insured by Medicaid. A large portion (44.1%) of pediatric severe sepsis ED visits occurred in the South census region. ED length of stay was over 3 h, and 16.5% were admitted to the hospital.

**Conclusion.** Nearly 100,000 children annually present to US EDs with severe sepsis. The findings of this study highlight the unique characteristics of children treated in the ED for severe sepsis.

## INTRODUCTION

Severe sepsis is the syndrome of systemic inflammatory response to microbial infection complicated by organ dysfunction. In adults, severe sepsis poses a major public health burden, resulting in over 750,000 hospitalizations, 500,000 Emergency Department (ED)

visits, and over 200,000 deaths in the United States (US) annually (*Angus et al., 2001*; *Wang et al., 2007*). The estimated cost of adult severe sepsis in the US is over $16 billion annually (*Angus et al., 2001*). Severe sepsis is also a prominent condition among children in the US. Watson et al. estimated that there are over 42,000 annual pediatric hospital admissions for severe sepsis in the US, with a mortality rate of 10.2% (*Watson & Carcillo, 2005*). Children admitted with severe sepsis had an average length of stay of 31 days at a cost of over $40,000, translating to an estimated annual US cost of $1.7 billion (*Watson et al., 2003*).

As in adults, early aggressive therapy of severe sepsis is associated with improved outcomes in children (*Carcillo, Davis & Zaritsky, 1991*; *de Oliveira, 2010*). In children with septic shock, Han et al. observed increases in mortality odds with delays in fluid and inotrope delivery (*Han et al., 2003*). The ED is an important component of pediatric sepsis care, presenting one of the earliest opportunities for recognition and treatment of the condition. However, the limited information regarding children receiving care for severe sepsis in US EDs presents a major gap in current sepsis knowledge. Data regarding number and types of patients, their characteristics, the therapies provided, and their outcomes could help to characterize the national burden of pediatric severe sepsis to US EDs and could inform strategies to optimize the delivery of pediatric severe sepsis care nationally.

In this study we sought to determine the characteristics of ED visits by children for severe sepsis in the US.

## METHODS

### Study design
Written approval from the University of Alabama at Birmingham Institutional Review Board was obtained for this study. This study was a cross-sectional analysis using the National Hospital Ambulatory Medical Care Survey (NHAMCS).

### Data source
Operated by the National Center for Health Statistics, NHAMCS is a national probability sample characterizing ED visits at hospitals across the US (NCHS). Standard sampling methods have been utilized by NHAMCS since its inception in 1973. Briefly, the study uses a four-stage probability design, sampling geographically defined areas, hospitals within these areas, emergency service areas within the emergency departments of the hospitals, and patient visits to the emergency services areas. For selected ED facilities, NHAMCS systematically selects records for approximately 100 ED visits in an assigned four-week period. The National Center for Health Statistics (NCHS) works with each hospital to abstract clinical data from selected charts. Each observation in NHAMCS reflects an individual ED visit; the data do not identify individual persons. The NHAMCS data collected in this manner have been used in over 500 publications (*McCaig & Burt, 2012*). For this study, we used NHAMCS public-use data for the nine-year period 2001–2009.

## Outcomes

The primary outcome was severe sepsis, defined as presentation to the ED with both (1) infection and (2) evidence of organ dysfunction (*Goldstein, Giroir & Randolph, 2005*). This approach has been widely used in a range of studies characterizing sepsis epidemiology (*Angus et al., 2001*; *Wang et al., 2007*; *Weycker et al., 2003*). We used a combination of triage vital signs and ED diagnoses as indicators of an infection and/or organ dysfunction. For each ED visit, NHAMCS reported up to three ED diagnoses using the *International Classification of Diseases, Ninth Revision* (ICD-9). We did not use blood culture results to define sepsis.

As described in prior studies, we identified infections using a taxonomy of ICD-9 codes developed by Angus et al. (*Angus et al., 2001*; *Wang et al., 2007*). However, because NHAMCS reported only three ED diagnoses, there was potential for under-reporting of infections. Therefore, consistent with prior efforts involving adult ED patients, we also defined the presence of fever or hypothermia (temperature $< 36°C$ or $\geq 38°C$) as indicators of an infection (*Goldstein, Giroir & Randolph, 2005*). The temperature was based upon initial measurement at triage; repeat measurements were not available. NHAMCS did not report the route of temperature measurement.

To identify organ dysfunction, we similarly followed the ICD-9 taxonomy of Angus et al. but added additional criteria (*Angus et al., 2001*). Additional ICD-9 codes representing organ dysfunction included 518.8 (respiratory failure), 786.03 (apnea), 799.1 (respiratory arrest), 990.90 (systemic inflammatory response syndrome/SIRS-not otherwise specified) and 995.92 (SIRS-infection with organ dysfunction) (*Angus et al., 2001*; *Wang et al., 2007*). We added ED endotracheal intubation as a form of organ dysfunction (respiratory failure). We also included ED triage hypotension an indicator of organ dysfunction (circulatory system failure), using international age-specific definitions for hypotension (*Gebara, 2005*; *Goldstein, Giroir & Randolph, 2005*) (Appendix S1). We opted not to use organ dysfunction definitions proposed by Goldstein et al. because of the unavailability of laboratory test values and physiologic measures in the NHAMCS data set (*Goldstein, Giroir & Randolph, 2005*).

## Covariates

Demographic characteristics included age, sex, race, ethnicity, hospital geographic region and population setting, month of visit and mode and time of arrival. Sex, race and ethnicity information was missing for some of the study population (0.86% missing sex information, 12.4% missing race and 17.5% missing ethnicity). Therefore, these characteristics were assessed using imputed variables provided by NHAMCS (NCHS). Geographic regions were based upon US Census Regions (Northeast, Midwest, West and South). Population setting consisted of hospitals in Metropolitan Statistical Areas (MSA) and non-MSAs. Month of visit was categorized as fall (September/October/November), winter (December/January/February), spring (March/April/May) or summer (June/July/August). Mode of arrival was categorized as arrival by ambulance vs. other. We divided time of arrival into 8-h intervals: 7 am–3 pm, 3 pm–11 pm, and 11 pm–7 am. We also examined

length of ED stay, admission to the hospital, admission destination and hospital discharge status. Initial ED vital signs were also examined, including heart rate (beats/min), systolic blood pressure (mmHg), and temperature (degrees Fahrenheit).

## Data analysis

Using descriptive statistics, we determined the annual number and characteristics of pediatric severe sepsis cases. We divided the patient cohort into five age subgroups to reflect definitions for neonates (0–30 days), infants (31 days–1 year), toddler and preschool (2–5 years), school age children (6–12 years), and adolescents (13–18 years) (*Goldstein, Giroir & Randolph, 2005*). We incorporated sampling design and weight variables to calculate nationally weighted estimates and their corresponding 95% confidence intervals. Performing the analysis in this manner allowed for valid estimation of all statistics and variances, accounting for the complex survey design.

The NCHS considers estimates based upon fewer than 30 raw observations (or those with greater than 30% relative standard error) to be unreliable. Therefore, we collapsed subcategories to ensure that table cells contained a sufficient number of raw observations. For example, we combined hematological, neurologic, hepatic and other organ dysfunctions into a single category. We used ultimate cluster design (single stage sampling) in variance and 95% confidence interval calculations, utilizing "masked" stratum and primary sampling unit identifiers provided with the NHAMCS public-use data set (NCHS). Prior efforts have demonstrated that variance estimates using these methods are conservative (*Hing et al., 2003*). We conducted all analyses using Stata v.12 (Stata, College Station, TX).

## RESULTS

An estimated 1.05 billion ED visits occurred in the U.S. during the nine-year study period (2001–2009), of which 253.4 million (24.1%) involved children <18 years of age, translating to approximately 28.2 million pediatric emergency visits annually (Table 1). One in three (34.3%) pediatric ED visits involved children with signs of infection, and approximately 1% involved organ dysfunction. The presence of a fever and an ED diagnosis for respiratory infection were the most common indicators of infection (Table 2). Hypotension and respiratory failure were the most common indicators of organ dysfunction.

Severe sepsis was present in 95,055 cases annually, representing 0.34% (95% CI: 0.29–0.39%) of all ED visits by children. The annual number of ED visits for pediatric severe sepsis declined slightly during 2001–2009 (Fig. 1).

Children between ages 31 days–1 year comprised the largest proportion of severe sepsis cases (Table 3). Most patients were white and non-Hispanic. Most pediatric severe sepsis ED visits occurred during winter months (37.4%). Almost half (45%) of pediatric severe sepsis patients arrived between 3:00 PM and 10:59 PM. Only 11.1% arrived by ambulance. Most pediatric severe sepsis visits occurred at hospitals in Metropolitan Service Areas. While most severe sepsis ED visits occurred in the South census region (40%), the proportions of severe sepsis in each region were similar ($p = 0.40$). Over half of pediatric

**Table 1 Emergency department (ED) visits by pediatric patients for severe sepsis.**

| Variable | No. of raw observations | Total ED visits 2001–2009 (estimated 1000s) | Annual ED visits (estimated 1000s) | Percentage of pediatric ED visits (95% CI) |
|---|---|---|---|---|
| Total ED visits | 322,745 | 1,052,914 | 116,990 | NA |
| Pediatric (age <18 years) ED visits 2001–2009 | 76,444 | 253,417 | 28,157 | NA |
| ED visits with "Infection" | 26,059 | 86,906 | 9,656 | 34.3 (33.5–35.1) |
| ED visits with "Organ Dysfunction" | 794 | 2,640 | 293 | 1.04 (0.94–1.15) |
| ED visits with severe sepsis ("Infection" + "Organ Dysfunction") | 266 | 855 | 95 | 0.34 (0.29–0.39) |

**Notes.**

CI = Confidence Interval; NA = Not applicable. "Infection" was defined as the presence of fever or hypothermia ($T < 36$ or $\geq 38°C$), or an ICD-9 diagnosis code for infection. "Organ Dysfunction" was defined as the presence of hypotension (based upon age-appropriate ED systolic blood pressure), provision of endotracheal intubation, or an ICD-9 diagnosis code for organ dysfunction.

**Table 2 Underlying infections and organ dysfunctions of Emergency Department visits by pediatric patients with suspected severe sepsis (age <18 yrs).**

| Variable | Number of raw observations | Total number of ED severe sepsis 2001–2009 (estimated 1000s) | Annual number of ED severe sepsis (estimated 1000s) | Percentage of ED severe sepsis (95% CI) |
|---|---|---|---|---|
| | | $N = 855,493$ | $N = 95,055$ | |
| **"Infection"** | | | | |
| Fever (<96.8 or ≥100.4F) on ED triage | 154 | 489 | 54 | 58.4 (50.6–65.8) |
| ED ICD-9 defined infection of respiratory system | 109 | 356 | 40 | 41.6 (33.5–50.2) |
| ED ICD-9 defined infection of genitourinary, digestive, nervous, musculoskeletal, or circulatory systems or other infections (parasitic, skin, or associated with pregnancy) | 50 | 183 | 20 | 21.4 (15.4–28.8) |
| **"Organ Dysfunction"** | | | | |
| Hypotension on ED triage | 201 | 632 | 70 | 73.8 (65.9–80.5) |
| ED ICD-9 defined respiratory dysfunction | 55 | 187 | 21 | 21.9 (15.8–29.4) |
| ED cardiovascular, hematological, neurologic, hepatic, or other organ dysfunction | 16[a] | 56[a] | 6[a] | 6.6 (3.2–13.1)[a] |

**Notes.**

CI = confidence interval; SBP = systolic blood pressure. "Infection" was defined as the presence of fever or hypothermia ($T < 36$ or $\geq 38°C$), or an ICD-9 diagnosis code for infection. "Organ Dysfunction" was defined as the presence of hypotension (based upon age-appropriate ED systolic blood pressure), provision of endotracheal intubation, or an ICD-9 diagnosis code for organ dysfunction.

[a] Estimate based upon <30 raw observations, considered unreliable by the National Center for Health Statistics.

**Table 3** Characteristics of Emergency Department pediatric patients presenting with severe sepsis.

| Variable | Number of raw observations | Total number of ED severe sepsis (2001–2009) (estimated 1000s) | Annual number of ED severe sepsis (estimated 1000s) | Percentage of ED severe sepsis (95% CI) |
|---|---|---|---|---|
| | | $N = 855{,}493$ | $N = 95{,}055$ | |
| Age | | | | |
| 0–30 Days | 16[a] | 63[a] | 7[a] | 7.3 (4.1–12.7)[a] |
| 31 Days–1 Year | 79 | 274 | 30 | 32.1 (25.0–40.0) |
| 2–5 Years | 46 | 133 | 15 | 15.5 (10.8–21.7) |
| 6–12 Years | 65 | 180 | 20 | 21.0 (16.3–26.6) |
| 13–18 Years | 60 | 207 | 23 | 24.2 (18.3–31.3) |
| Sex | | | | |
| Male | 120 | 374 | 42 | 43.7 (36.2–51.5) |
| Female | 146 | 481 | 53 | 56.3 (48.5–63.8) |
| Race | | | | |
| White | 189 | 576 | 64 | 67.3 (59.3–74.5) |
| Black | 61 | 231 | 26 | 27.0 (20.5–34.6) |
| Other | 16[a] | 49[a] | 5[a] | 5.7 (3.0–10.8)[a] |
| Ethnicity[b] | | | | |
| Hispanic | 52 | 155 | 17 | 23.8 (17.1–32.2) |
| Not Hispanic | 148 | 494 | 55 | 76.2 (67.8–82.9) |
| Season | | | | |
| Fall (Sept, Oct, Nov) | 65 | 204 | 23 | 23.9 (17.9–31.1) |
| Winter (Dec, Jan, Feb) | 91 | 320 | 36 | 37.4 (29.6–46.0) |
| Spring (Mar, Apr, May) | 61 | 199 | 22 | 23.2 (16.7–31.3) |
| Summer (Jun, Jul, Aug) | 49 | 133 | 15 | 15.5 (10.5–22.4) |
| Arrival Time | | | | |
| 7:00AM–2:59PM | 100 | 312 | 35 | 36.5 (29.2–44.5) |
| 3:00PM–10:59PM | 116 | 390 | 43 | 45.7 (38.6–53.0) |
| 11:00PM–6:59AM | 49 | 152 | 17 | 17.8 (12.4–24.8) |
| Arrival By Ambulance[b] | | | | |
| Yes | 22[a] | 66[a] | 7[a] | 11.1 (6.6–18.0)[a] |
| No | 164 | 529 | 59 | 88.9 (82.0–93.4) |
| Region | | | | |
| Northeast | 62 | 142 | 16 | 16.6 (12.0–22.5) |
| Midwest | 50 | 162 | 18 | 18.9 (13.2–26.4) |
| South | 96 | 377 | 42 | 44.1 (35.7–52.9) |
| West | 58 | 174 | 19 | 20.3 (14.1–28.4) |
| Hospital Population Setting | | | | |
| Metropolitan Statistical Area (MSA) | 223 | 651 | 72 | 76.1 (64.6–84.7) |
| Non-MSA | 43 | 205 | 23 | 23.9 (15.3–35.4) |
| Payor Type | | | | |
| Private Insurance | 94 | 274 | 30 | 32.2 (25.9–39.2) |
| Medicaid | 133 | 443 | 49 | 51.9 (44.0–59.8) |
| Self-Pay | 17[a] | 45[a] | 5[a] | 5.3 (3.1–9.0)[a] |

Table 3 (*continued*)

| Variable | Number of raw observations | Total number of ED severe sepsis (2001–2009) (estimated 1000s) | Annual number of ED severe sepsis (estimated 1000s) | Percentage of ED severe sepsis (95% CI) |
|---|---|---|---|---|
| Other | 20[a] | 90[a] | 10[a] | 10.6 (6.3–17.3)[a] |
| Admitted to Hospital | | | | |
| Yes | 50 | 141 | 16 | 16.5 (11.3–23.5) |
| No | 216 | 715 | 79 | 83.5 (76.5–88.8) |
| Admitted to Critical Care Unit | | | | |
| Yes | 13[a] | 27[a] | 3[a] | 3.1 (1.4–6.8)[a] |
| No | 253 | 829 | 92 | 96.9 (93.2–98.6) |
| Length of ED Stay (Hours) (95% CI) | | | | 3.3 (2.6–4.0) |

**Notes.**

ED = emergency department, CI = confidence interval.

[a] Estimate based on <30 raw observations, considered unreliable by the National Center for Health Statistics.

[b] Data available for 2003–2009 only.

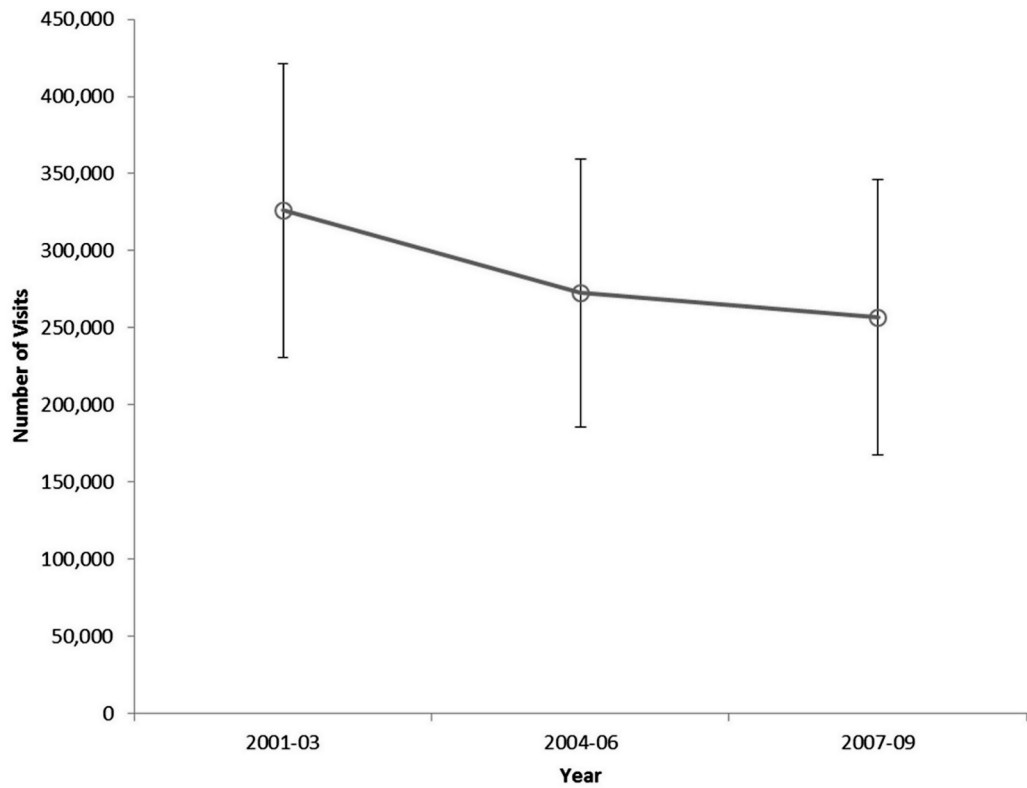

**Figure 1 Number of pediatric severe sepsis Emergency Department visits by three-year interval, United States, 2001–2009.**

severe sepsis cases were self-pay or insured by Medicaid. Pediatric severe sepsis patients spent over three hours in the ED. In the NHAMCS data set, 16.5% of pediatric severe sepsis ED visits resulted in hospital admission, and 3.1% were admitted to a critical care unit.

## DISCUSSION

Our study suggests that there are almost 100,000 ED visits for pediatric severe sepsis each year in the US. Severe sepsis is associated with significant morbidity and mortality, and early aggressive therapy may help to improve outcomes from this condition (*Rivers et al., 2001*). Our study illustrates the potential magnitude of ED pediatric severe sepsis population in the United States. While there are limitations inherent to the NHAMCS data set, our observations offer the best available and most current cross-sectional perspectives of pediatric severe sepsis in the US.

Our findings shed light on the characteristics of children requiring severe sepsis care in the ED. The age distribution of severe sepsis was bimodal, with most cases occurring in children ages 31 days–1 year and 13–18 years. Only 11.1% arrived by EMS, indicating opportunities to expand the involvement of EMS in the care and transportation of these children (*Wang et al., 2010*). The insurance payor for over half of severe sepsis cases was Medicaid or self-pay, highlighting the prevalence of this condition among those with lower socioeconomic status. While not indicated by these data, barriers to timely primary or preventive care may have amplified the number of ED visits by this subgroup.

Using 1995 hospital discharge data from seven states, Watson et al. estimated that there were 42,364 annual hospitalizations for pediatric severe sepsis in the US (*Watson & Carcillo, 2005*). Our contrasting study focuses on children presenting to the ED, which is often the location for initial detection and management of sepsis. In contrast to the Watson study, we observed only 16,000 annual pediatric severe sepsis hospital admissions from the ED as well as low rates of critical care unit admission. We urge caution when making inferences from these unexpected observations, which are clearly discordant with Watson's prior work. There are many potential explanations for the disagreement between Watson's and our study. Watson's study included low birth weight neonates in neonatal intensive care units, which is not representative of the general ED population. The majority of cases in the Watson series may reflect children who developed severe sepsis during later points of hospitalization. Many pediatric patients in our series may have received successful initial treatment with sufficient recovery for discharge home from the ED. The data from the Watson study are also almost 18 years old, and may not represent current patient characteristics or clinical practices.

However, another potential explanation is that NHAMCS coding or data abstraction errors unique to the data set or this population may have resulted in underestimates of the number of admissions. Recent studies highlight concerns with NHAMCS data pertaining to hospital admission status or destination; these factors clearly apply to our analysis (*Cooper, 2012*; *Green, 2013*; *McCaig & Burt, 2012*). We note that we did not perform an independent validation of the NHAMCS dataset for this particular analysis. However, a proper validation study would be logistically challenging, requiring manual review of medical records from multiple EDs. We emphasize that these inconsistencies related to ED disposition should not distract from the main premise of our study – that the annual number of US ED visits for severe sepsis is large and diverse. Our study is intended to

be hypothesis generating and to provide foundation knowledge for characterizing the collective burden of pediatric severe sepsis upon the US emergency care system.

The findings of our study highlight important issues in pediatric sepsis care. The care of the pediatric severe sepsis patient is often complex, involving early recognition and aggressive resuscitative care. Outcomes for complex medical conditions are often better at centers that care for high volumes of patients; for example, trauma centers that care for larger volumes of injured patients report improved survival (*Nathens et al., 2001*). The National Emergency Department Inventory (NEDI-USA) estimated that there were 4,874 EDs in the US in 2007 (NEDI-USA). Based upon this figure and our finding of approximately 100,000 annual ED visits for pediatric severe sepsis, one would expect a typical ED to care for approximately 20 pediatric severe sepsis cases each year. If additional studies were to confirm a minimum experience threshold, clinicians and policymakers might entertain alternate pediatric sepsis care strategies. For example, providers might triage pediatric severe sepsis cases to specialized pediatric EDs with expertise in sepsis resuscitation. Clinicians may also devise ways to improve severe sepsis recognition, such as through the use of point-of-care lactate measurements (*Levy et al., 2010*; *Perel, 2008*; *Rivers et al., 2001*). We emphasize that our study does not indicate the effectiveness of such approaches. Rather, our study highlights the sizable pediatric population that could benefit from optimized severe sepsis detection and management.

## LIMITATIONS

NHAMCS is retrospective in nature and uses a probability sample design. However, the methodology of the NHAMCS study is rigorous, and the data set has been widely used in similar analyses for over 15 years. Due to the absence of individual identifiers, we could not estimate rates of ED re-visits. Because NHAMCS collects only three diagnoses per patient, we may have missed additional relevant conditions that were not reported by data abstractors. Abstractors may have also varied in the selection of ED diagnoses; it is unclear whether this bias would have resulted in under- or over-estimates of the number of severe sepsis cases. Our focus on children resulted in a relatively modest sample size, limiting inferences for smaller subgroups. Most notably, while representing an important sepsis subset, our ability to characterize neonates was limited. We note that given the current data we could not correct ages for those born prematurely.

As discussed previously, the identification of severe sepsis using diagnostic codes and vital signs has not been prospectively validated with the current data set, an effort that would be logistically complex. However, the strategy has been widely used by other studies, including those using NHAMCS (*Wang et al., 2007*). While Martin, et al. proposed a set of ICD-9 codes corresponding to sepsis, we focused on severe sepsis (encompassing the combination of infection and organ dysfunction codes) in this study because of the unclear coding practices for children (*Martin et al., 2003*). We note that the definition of sepsis is controversial, and the standards used in this study reflect common practice used in a range of prior studies (*Angus et al., 2001*; *Wang et al., 2007*).

Our analysis was based upon initial vital signs and did not preclude those receiving prior treatment (for example, administrative of antipyretics prior to ED arrival) or those decompensating at later time points in the ED. We were unable to disentangle infectious from non-infectious causes for fever or hypotension in the current data set. We also focused on children presenting to the ED with severe sepsis, not children developing sepsis as a result of hospitalization for other medical conditions. We could not identify conditions such as malignancies or congenital deficiencies that often predispose children to sepsis (*Orange, 2005*). Because of the small relative number of raw observations, we could not provide detailed insights on ED interventions or hospital outcome. Our ability to describe secular trends in sepsis was also limited.

## CONCLUSION

Nearly 100,000 children annually present to US EDs with severe sepsis. The findings of this study highlight the unique characteristics of children treated in the ED for severe sepsis.

### Funding

The authors have no external funding sources to disclose.

### Competing Interests

Dr. Henry Wang is an Academic Editor for PeerJ.

### Author Contributions

- Sara Singhal conceived and designed the experiments, wrote the paper.
- Mathias W. Allen conceived and designed the experiments, performed the experiments, analyzed the data.
- John-Ryan McAnnally performed the experiments, analyzed the data.
- Kenneth S. Smith wrote the paper.
- John P. Donnelly performed the experiments, analyzed the data, wrote the paper.
- Henry E. Wang conceived and designed the experiments, performed the experiments, analyzed the data, wrote the paper.

### Human Ethics

The following information was supplied relating to ethical approvals (i.e. approving body and any reference numbers):

The Institutional Review Board of the University of Alabama at Birmingham approved this study: UAB IRB Protocol N090512004.

### Supplemental Information

Supplemental information for this article can be found online at http://dx.doi.org/10.7717/peerj.79.

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
