# Peer review of "National estimates of emergency department visits for pediatric severe sepsis in the United States"

_PeerJ, doi:10.7717/peerj.79_

## Round 0.1 · original submission · Major Revisions

Dear Dr Singhal and colleagues,

Thank you for submitting your article on ED visits for pediatric severe sepsis to PeerJ. Two expert reviewers have assessed your manuscript, and have raised some significant issues. I would encourage you to submit a revision. Please note I can make no guarantee of acceptance after revision. Your revision will be peer-reviewed once again before a decision on publication is made.

In your response, please ensure you address each of the points made by the Reviewers.

Thank you again for submitting your work to PeerJ.

Philip Jones, MD MSc FRCPC
* * *
Reviewer 1 ·

Basic reporting

Page 4, Data Source – The description of the data does not allow one to determine if duplicate patients are included. For e.g. is it possible that some patients from ERs in smaller centres may have been transferred to ERs in a larger more specialized centre, and therefore, have been counted twice?

Page 5, Outcomes, Line 85-86 – How was “serious infection” defined?

Page 5, Outcomes – “We used a combination of triage signs and ED diagnoses as indicators of an infection and/or organ dysfunction.” It is not clear what vital signs criteria were used.

Page 5, Outcomes – There is no discussion as to what proportion of sepsis diagnoses were supported by positive blood cultures.

Page 7, Data Analysis, Line 128 – Subcategories were collapsed but it is not described how this was done.

Page 8, Results, Line 141 – In the results section severe sepsis is defined as the presence of infection and organ dysfunction, which contradicts the definition in the Outcomes section (Page 5) where a “serious” infection is required.

Experimental design

Page 5, Outcomes – “we also defined the presence of fever or hypothermia (triage temperature ….) as indicators of infection”. However, there are other causes of fever in addition to infection and this possibility is not discussed.

Page 13, Limitations, Line 223-224 – “the methodology has not been prospectively validated with the current data set.” I see this limitation as a major one.

Validity of the findings

Page 11, Discussion, Lines 185-186 – “we observed … as well as hospital mortality.” I was unable to find any results describing mortality.

Page 11, Discussion, Lines 188-190 – “However, our estimate …is consistent relative to Watson’s estimate of total annual severe sepsis hospitalizations.” I do not see this relationship.

Page 11, Discussion, Line 195-198 – “Based upon the National Emergency Department Inventory estimate … for only 20 pediatric severe sepsis cases annually, a relatively small number”. I am not sure this is a valid calculation.

Page 15, Conclusion, Line 237-239 – The findings of this study highlights … the challenges of providing timely and expert pediatric sepsis care.” I do not think that this study addresses these challenges and therefore this conclusion cannot be made based on its results.

·

Basic reporting

The authors submit a very interesting and well written paper characterizing pediatric patients identified as having severe sepsis who have visited a United States Emergency Department during the period 2001-2009. The data source, the National Hospital Ambulatory Medical Care Survey (NHAMCS), is a well established source of national hospital statistics data. While not listed as a required section header in the standard PeerJ template, the inclusion of a separate limitations section is appropriate.

Experimental design

An original research question is presented and clearly defined. The research question is meaningful as this seeks to characterize the burden of pediatric severe sepsis cases presenting to Emergency Departments in the United States, which has important clinical as well as healthcare resource utilization implications. The experimental design is appropriate to the question and the methods are well described. Importantly, characteristics of the source data/database are described in an appropriate level of detail for the reader to understand the strengths and limitations inherent in its use. Ethics approval for the research was obtained from the author's institution.

Validity of the findings

The authors appropriately utilize descriptive statistics to present their data. I am unfamiliar with the reported method of variance and 95% CI determination, though a reference for this is provided. Methods for case definition/identification are clearly described e.g. a case of severe sepsis had 1) serious infection (based on previously established ICD-9 code taxonomy) OR the presence of triage fever or hypothermia (T<96.8 F or >/= 100.4 F) AND 2) Organ dysfunction (based on previously established ICD-9 code taxonomy) as well as the inclusion of additional ICD-9 codes for respiratory failure, apnea, respiratory arrest, SIRS/SIRS-NOS, and SIRS-infection with organ dysfunction. Endotracheal intubation was also included as a surrogate for respiratory failure. The presence of triage hypotension based on age-related norms was used to identify cases with objective findings of circulatory failure although they do appropriately distinguish these cases from those with ICD-9 coded circulatory dysfunction in Table 2. The authors acknowledge that due to methods of NHAMCS data abstraction, that some cases may have been missed, however the authors’ inclusion of additional ICD-9 codes has sought to mitigate this. Comment should be provided in the paper regarding how case identification methods ensured that double-counting of cases did not occur.

As I’m sure the authors are aware, their case definition of severe sepsis differs somewhat from the definition provided in the 'International pediatric consensus conference: definitions for sepsis and organ dysfunction in pediatrics' (Goldstein B, et al. 2005). I acknowledge that this is due in part to the limitations in the data, but this should be specifically addressed, and I would suggest comparing and contrasting their case definitions to that provided in the Goldstein paper. For example, to meet criteria for severe sepsis according to consensus conference definitions, patients should meet criteria for sepsis AND have cardiovascular dysfunction OR acute respiratory distress syndrome (ARDS) OR two or more organ dysfunctions (these are all specifically characterized in Table 4 of the Goldstein paper). I suspect that based on the authors’ definitions, a number of identified cases meet the criteria for sepsis but not necessarily severe sepsis e.g. cases presenting with fever and respiratory distress but not actually meeting criteria for ARDS or requiring mechanical ventilation – a situation common among children presenting to the Emergency Department and supported by the increased preponderance of cases with 'severe sepsis' during the winter months. One way for readers to be able to directly compare and contrast the differences between the consensus conference definitions and those used in the authors' paper would be to include a table with the authors’ criteria on one side and the consensus conference definitions on the other.

Another issue which the authors should speak to is the high percentage of patients identified to have had hypotension at triage (73% of those with organ dysfunction), yet a minority of cases identified in this study were ultimately admitted to hospital. Given that hypotension in children is recognized as a serious and typically late finding in the course of illness, can the authors’ provide some explanation for this perplexing finding? Is it possible that some of these cases of ‘hypotension’ actually represent situations of ascertainment bias where children presenting (particularly to non-pediatric EDs) may have their blood pressure checked with a cuff of the wrong size? Or do some of these cases actually represent situations of acute hypovolemia in the setting of gastroenteritis – another common ED presentation – which may have been addressed and corrected in the ED setting and ultimately resulted in a disposition home from the ED?

Additional comments

Line 56-57: It would be helpful to readers to clearly link the adult vs. the pediatric citations to the related statements. This (line 56-57) sentence is a good example, and reads like all of the citations refer to pediatric outcomes, whereas only the Carcillo and de Oliveria papers do.
Line 58: Perhaps best described as increased mortality odds.
Line 85-86: The authors have used 'infection' and 'serious infection' at different points in the paper. Based on consensus conference definitions, suggest for consistency to keep this as simply 'infection' - unless they excluded some forms of infection from their case definition.
Line 95: Temperatures should be reported in Celsius to correspond with the cut-offs reported in the international pediatric consensus conference, although Farenheight values may also be reported and would be of interest/utility to many U.S. readers.
Line 136: Results - General comment - the tables are cited out of order and perhaps the organization of the content reported in the Results section could be improved. For example, Line 150-153 may be better moved up to closer to the beginning of this section.
Line 141: 'Defined as the presence of infection and organ dysfunction' - the initial part of this sentence speaks to methods which are already described in that section. Suggest start with 'Severe sepsis was present...'
Line 146-147: This sentence in the results section also duplicates definitions reported in the methods and should be eliminated.
Line 159: Would be interesting to report mortality data if available.
Line 171: change 'includes' to 'included'
Line 173: change 'of' to 'from'; remove 'the most'
Line 180: suggest change prominence to incidence
Line 207: Another example of where it would be clearer to link the adult citations to the first part of the sentence e.g. include following the comma
Line 210: Limitations. At some point in the paper the authors should mention that the ages reported are chronological rather than corrected ages, which is relevant to their age-based classification in terms of how formerly premature patients are classified. Also, as mentioned under outcomes, the authors note the limitation of temperature location not being consistent. It may also be worth mentioning as a limitation that the absence of triage fever does not preclude fever having been an important part of the presenting illness e.g. parents frequently administer antipyretics to children at home prior to attending to the ED, where a temperature may then not be detected.

Tables: I suggest a change to the language used in Tables 1, 2, and 3 to avoid confusion. 'Pediatric Emergency Department visits' to many readers will suggest that the patient was seen in a specialized PED setting. I believe that what the authors are trying to indicate is that these are Pediatric visits to an Emergency Department setting - an important distinction. Applies to both table titles and descriptions.

Figures: The included figure is appropriate and informative.

---

## Round 0.2 · Minor Revisions

Dear Dr Singhal and colleagues,

Thank you for submitting an excellent revision which incorporates the changes and suggestions made by the Reviewers.

One of the Reviewers (Dr Parker) is happy with the manuscript as-is, but, as you can see, the other Reviewer requests that additional, frank discussion be included on the limitations of using the NHAMCS dataset for the inferences made in your manuscript. I tend to agree with the Reviewer that, given the importance of the validity of the dataset to the validity of your findings, it may be prudent to include a small amount of additional discussion as to why, given the lack of dataset validation in this context, your findings may still be interpretable. This may be as simple as stating that this work is targeted towards hypothesis generation rather than for effecting change. However, I will leave this to you.

I have to admit I am also surprised by the low proportion of children with severe sepsis admitted to hospital. Brief additional comments on this topic would be encouraged.

If you would please address the Reviewer's comments about dataset validity and the low admission rate for patients with "severe sepsis" in a revision, I think your paper will be ready for publication.

Thank you again for submitting an excellent revision.

Reviewer 1 ·

Basic reporting

No new comments.

Experimental design

No new comments.

Validity of the findings

The authors have addressed most of my concerns. However, I still remain concerned regarding the statement that “the identification of severe sepsis using diagnostic codes and vital signs has not been prospectively validated with the current data set”. This methodology is the basis of their study and if it has not been validated it calls into question their findings and conclusions. Justification that the strategy has been widely used by other studies is not adequate. More frank discussion of this limitation and the effect it may have on the results is required.

Page 9, Results, Line 162 – “16.5% of pediatric severe sepsis ED visits resulted in hospital admission” and “Many pediatric patients in our series may have received successful initial treatment with sufficient recovery for discharge home from the ED”. This finding is surprising to me as I find it difficult to believe that the majority of patients with severe sepsis did not require hospital admission as I would assume that if the patients were indeed severely septic that they would require hospital admission. This finding makes me question whether a substantial proportion of the cases that they classified as severe sepsis based on their criteria were not actually cases of severe sepsis.

·

Basic reporting

The paper reads very well.

Experimental design

No further comment.

Validity of the findings

The authors have addressed the comments from my initial review to the best of their ability, given the limitations of the data.

Additional comments

The authors should be congratulated for taking on this difficult research question. Their paper adds to the existing literature on pediatric sepsis epidemiology and should prompt some good discussion! The reported findings raise a number of interesting questions and speak to the need for further research in this area.

---

## Round 0.3 · accepted · Accept

Dear Dr Singhal and colleagues,

Thank you for submitting a second revision. I think your nuanced edits have clearly emphasized the important limitations of this study, while still adequately demonstrating the manuscript's thought-provoking conclusions. I am very happy to now accept your paper for publication in PeerJ.

I think your paper will be of significant interest to PeerJ readers, and I look forward to seeing more work from your research team.

Sincerely,

Philip M Jones, MD MSc FRCPC